# Prevalence of alcohol use by gender and HIV status in rural Uganda

Adriane Wynn[1], Katelyn M. Sileo[2], Katherine Schmarje Crockett[3,4], Rose Naigino[3,4], Michael Ediau[3,4], Rhoda K. Wanyenze[5], Noah Kiwanuka[6], Natasha K. Martin[1], Susan M. Kiene[3,5] *

1 Division of Infectious Diseases and Global Public Health, University of California, San Diego, La Jolla, California, United States of America, 2 Department of Public Health, The University of Texas at San Antonio, San Antonio, Texas, United States of America, 3 Division of Epidemiology and Biostatistics, School of Public Health, San Diego State University, San Diego, California, United States of America, 4 The Herbert Wertheim School of Public Health and Human Longevity Science, University of California San Diego, La Jolla, California, United States of America, 5 Department of Disease Control and Environmental Health, Makerere University School of Public Health, Kampala, Uganda, 6 Epidemiology and Biostatistics Department, at Makerere University School of Public Health, Kampala, Uganda

* skiene@sdsu.edu

Data Availability Statement: All relevant data are within the paper and its Supporting information files.

Funding: This study was funded through The National Institute of Mental Health, National

## Abstract

### Background

Alcohol use is a major contributor to mortality and morbidity worldwide. Uganda has a high level of alcohol use per capita. Compared to men, women are less likely to consume alcohol globally; however, women who drink have increased risks for co-occurring conditions, including depression, intimate partner violence, and HIV. This study assessed the prevalence of alcohol use and correlates of harmful alcohol use by gender and HIV status in rural Uganda.

### Methods

We used cross-sectional data from a study among women and men aged 15–59 residing in rural, central Uganda and accepting home-based HIV testing (Nov 2017 to Dec 2020). We estimated the prevalence of levels of alcohol use (categorized as no alcohol use (score 0), low (score 1–3 for men; 1–2 for women), medium (score 4–5 for men; 3–5 for women), high (score 6–7), and very-high (score 8–12) use with the AUDIT-C), stratified by gender and HIV status. We assessed correlates of harmful alcohol use using multivariable logistic regression models for women and men.

### Results

Among 18,460 participants, 67% (95% CI: 66–67%) reported no alcohol use, 16% (95% CI: 16–17%) reported low, 5% (95% CI: 4.8–5%) reported medium, 5% (95% CI: 4–5%) reported high, and 3% (95% CI: 2.8–3) reported very high alcohol use. Compared to women, men were more likely to report alcohol use (Chi-squared p-value<0.0001). People diagnosed with HIV (both newly diagnosed and previously aware of their status prior to home-based HIV testing) were more likely to report low, medium, high, and very high alcohol

Institutes of Health, award R01MH106391 to SK and RKW. AW was supported by a National Institute on Alcohol Abuse and Alcoholism K01 Career Development Award (K01AA027733). The funders had no role in the study design, data collection and analysis, decision to publish or preparation of the manuscript.

**Competing interests:** The authors have declared that no competing interests exist.

use compared to those who were HIV negative (Chi-squared p-value<0.0001). Among women, those who were newly diagnosed were more likely report alcohol use, compared to those who were HIV negative. In multivariable models, being newly diagnosed with HIV (compared to HIV negative) increased the odds of harmful alcohol use among women, but not men.

## Conclusion

While alcohol use was higher among men and people living with HIV, being newly diagnosed with HIV had a stronger relationship with harmful alcohol use among women than men. More research is needed to understand how alcohol use may increase the risks of HIV acquisition among women and to identify gender-responsive services to address harmful alcohol use and increase access to HIV testing and linkage to care for women who use harmful levels of alcohol.

## Introduction

Although alcohol use is common globally, higher levels of use can have detrimental health and social consequences [1]. Alcohol use is a major contributor to mortality and morbidity worldwide [2], with links to over 200 communicable and non-communicable diseases and injuries [3]. The World Health Organization (WHO) Africa Region bears the largest age-standardized alcohol-attributable burden of disease and injury [1].

Uganda has among the highest estimated levels of alcohol use per capita in the WHO Africa Region [1]. According to WHO estimates, a substantial proportion of the population in Uganda abstained from alcohol use in the past year (63.7%); however, among those who drank, 56.9% reported heavy episodic drinking (68.8% among men and 32.6% among women), which is defined as ≥60 grams of pure alcohol on at least one occasion at least once per month [1]. Further, the prevalence of alcohol use disorders (including dependence and harmful patterns of alcohol use) in the general population was 7.1% (12.4% among men and 1.9% among women) [1].

Alcohol use and related outcomes also vary by gender. Compared to men, women are less likely to consume alcohol and have lower risks for problematic alcohol use and alcohol-related harms worldwide [4]. However, women who drink have higher risks for certain alcohol related problems, compared to men. Women who misuse alcohol are more likely to develop alcohol hepatitis [5], alcohol related heart disease,(3) and brain damage [6]. Studies have also found an association between drinking alcohol and breast cancer [7] and drinking during pregnancy increases risks for physical, cognitive, and behavioral problems in children [8]. Women face co-occurring conditions, including depression, intimate partner violence, and HIV [9]. Further, previous research in Russia and the United States found that, among people living with HIV, the gender gap in heavy alcohol use converges; however, this difference was not found in Uganda [10], where the incidence of HIV is higher among women compared to men [11,12]. While past research has assessed the prevalence of alcohol use among men and women [1,13] or among people living with HIV [14,15] in Uganda, few studies have estimated the prevalence stratified by both gender and HIV status. Such nuanced prevalence estimates are needed by researchers and policy-makers to understand the burden of alcohol use and to determine service needs.

This research aimed to estimate the prevalence of alcohol use with a multi-level measure of consumption (none, low, medium, high, and very high) stratified by both gender and HIV status using a large general population sample in rural, Central Uganda. We assessed correlates of harmful alcohol use using multivariable logistic regression models for women and men. We hypothesized that men and people living with HIV, particularly those newly diagnosed, would drink alcohol at higher levels than women and participants who were HIV negative. We also hypothesized that harmful alcohol use would have a stronger association with HIV among women compared to men.

## Methods

### Parent study

This study used cross-sectional baseline data from the PATH/Ekkubo matched-pair, two-arm, cluster randomized controlled trial in rural districts of Butambala, Mpigi, Mityana, and Gomba in central Uganda [16]. Matched villages were randomized to either the standard-of-care arm or the intervention arm, which included an enhanced linkage to HIV care intervention. Households within selected villages were included. Eligibility criteria for enrollment at baseline were: adults aged 18 to 59 years old or emancipated minors (e.g. individuals married or with children) aged 15 to 17 years, accepting HIV testing, speaking Luganda or English, and residing in a household in a study village. HIV counseling and testing was provided to all eligible and consenting participants. This study includes participants recruited between November 2017 and December 2020. As follow-up was requested for those enrolled in the trial, the Principal Investigators of the parent study had access to data that could identify individual participants.

### Measures

Data were collected from consenting participants using interviewer-administered, computer-based questionnaires, which included questions related to socio-demographic characteristics, alcohol use, and HIV status and care.

### Outcome measure: Alcohol use

Participants were asked about alcohol consumption using the 3-item Alcohol Use Disorders Identification Test (AUDIT-C) [17], which estimates the frequency and quantity of alcohol use. For analysis, AUDIT-C scores were categorized into no alcohol use (score 0), low (score 1–3 for men; 1–2 for women), medium (score 4–5 for men; 3–5 for women), high (score 6–7), and very-high (score 8–12) use. These levels have been identified in previous literature as demonstrating increased severity and adverse HIV care cascade outcomes, including reduced likelihood of ART, ART adherence, and viral suppression among populations in the United States and East Africa [15,18]. We assessed "harmful" alcohol use, which is measured as an AUDIT-C score of $\geq$ three for women and $\geq$ four for men and marks the threshold between no to low alcohol use and medium to very high use [17]. For questions related to alcohol use, participants were asked to consider beer, wine, and commonly consumed local alcoholic beverages marwa (malted millet), waragi (local gin), tonto (fermented bananas), or any beverage that contains alcohol; they were asked to exclude communion wine or wine received at church. To help with the accurate reporting of total drinks, participants were shown pictures of what constitutes a single drink using images of local beverages. Using the pictures as a guide, participants estimated the number of single drinks they had on a typical day when drinking and how often they consumed six or more single drinks on one occasion.

## Main exposures: HIV status and gender

HIV testing was performed by health workers using the WHO algorithm for generalized epidemics [19], which started with a finger stick capillary blood sample for an HIV rapid test assay (Alere/Abbott Laboratories, Chiba, Japan). For this analysis, we categorized those who reported as being previously diagnosed with HIV and confirmed positive as "known positive," those who reported no prior positive HIV test result or no prior HIV testing as "new positive," and those who tested negative as "HIV negative." Gender was self-reported as a dichotomous variable (male/female).

## Other correlates of harmful alcohol use

Socio-demographic information was collected, including gender, age, tribe, educational level, occupation, income, religion, marital/partnership status, and socioeconomic status (measured using a wealth index). Age was a continuous variable collapsed into three categories: 14–24, 25–34, and 35+ years. Marital status was assessed using the question "Are you currently married, separated, widowed, divorced, or have you never been married?" using the participant's own definition of marriage (measured as married and living together or married and separated most of the time, widowed, divorced, or never married). We collapsed responses into never married, divorced, widowed, and currently married. Highest education level was based on responses to the question "What is the highest level of school you completed?," which we collapsed into primary or less and secondary or greater. In the survey, religion was determined through the question "What is your religion?" with the following responses: Protestant (Anglican, Lutheran, Church of Uganda), Catholic, Muslim, Seventh Day Adventist, Saved/Pentecostal, None, and Other. We collapsed the categories into Protestant, Catholic, Muslim and Other (combined with None and Saved/Pentecostal) for analysis. Participants were asked, "What is your occupation?" We included the most common categories, which were peasant famer, fisherman, casual worker, and not employed; and we combined salaried workers with businesspersons. We also included occupations previously identified as increasing the risk for higher levels of alcohol use, such as restaurant and bar workers [13,20]. Data from the remaining occupations were too sparse for analysis. The wealth index used Demographic and Health Surveys (DHS) questions and procedures including factor analyses to create the index for the entire sample based on household characteristics, including roof and floor materials, and whether the household had electricity, television, or a sofa set [21].

Depressive symptoms were assessed using a modified 10-item version of the Center for Epidemiological Studies of Depression Short Form (CES-D-10) scale, an abbreviated version of the original 20-item scale [22]. The scale has been shown to be reliable in a rural Ugandan sample ($\alpha = 0.90$) [23]. Based on previous research showing a need for a higher cut-off score in populations in Sub-Saharan Africa [24], we used a threshold of 14 or higher to identify those at risk for depression.

## Statistical analysis

All analyses were conducted using STATA version 17. First, we examined participant characteristics by gender. Next, we estimated the prevalence of no, low, medium, high, and very high levels of alcohol use stratified by gender and HIV status, including new and known HIV positive. For each prevalence estimate, exact binomial 95% confidence intervals (CIs) were calculated and population weights were applied to balance the distribution of the sample by subcounty (sub-unit of a county within a district) relative to the subcounty population aged 18–59 using 2014 census data with projections for 2016 [25]. Chi-squared tests were used to compare alcohol use levels with gender and HIV status variables.

Finally, we assessed the relationship between HIV status (negative, newly diagnosed, previously diagnosed) and harmful alcohol use with multivariable logistic regressions and separate models were fit for each gender. Covariates were included in the model based on a conceptual framework developed by Shuper et al. [26], previous literature on of the relationship between alcohol use and HIV [13,20], and the availability of data from the PATH/Ekkubo baseline questionnaire. We also assessed an interaction between HIV status and depressive risk. Clustering could occur within villages and the intraclass correlations coefficient was 0.021, suggesting that 2.1% of the variance in harmful alcohol use was due to variations across village. Thus, to adjust for non-independence and heteroskedasticity of the errors, we used both the "robust" and "cluster" option in STATA.

## Ethics

The parent study received ethical approval by International Review Boards in the U.S. and Uganda: San Diego State University (# 2095100) and Makerere University School of Public Health in Uganda (#296). The Uganda National Council for Science and Technology approved the study. The University of California, San Diego Human Research Protections Program deemed these secondary analyses using de-identified data as exempt (#201316). Participants provided written informed consent for HIV testing and study participation.

## Results

### Sample characteristics

A total of 18,460 men and women were included in our analytical sample. Table 1 displays sociodemographic characteristics, CES-D-10 scores, and HIV status for the total population and stratified by gender. In our full sample, most participants were female (61%), married (63%), and had achieved primary or less education (67%). In terms of occupations, most were peasant farmers (41%), followed by salaried or business employees (16%), and part-time/day laborers (9%). The most common occupation reported in the "other" category was housewife/househusband (8%). Only 16 women reported being commercial sex workers. Most participants were in the lower levels of the wealth index, members of the Catholic religion, and part of the Muganda tribe. In terms of depression, 11% reported a CES-D-10 score of 14 or higher (at risk for depression). The overall HIV prevalence was 5.2% (1.6% were newly diagnosed and 4% were previously diagnosed). Compared to men, women were more likely to be younger, married, have a CES-D-10 score of 14 or higher, and be living with HIV.

### Prevalence of alcohol use by gender

In the sample, 67% (95% CI: 66–67%) reported no alcohol use, 16% (95% CI: 16–17%) reported low, 5% (95% CI: 4.8–5%) reported medium, 5% (95% CI: 4–5%) reported high, and 3% (95% CI: 2.8–3) reported very high alcohol use. Seventeen percent of participants reported harmful alcohol use (medium to very high use). Compared to men, women were more likely to abstain from alcohol use (59% vs. 72%), and more likely to report low (19% vs. 15%), high (7% vs. 3%), and very high (6% vs. 1%) alcohol use (Chi-squared p-value<0.0001). Men were also more likely to report harmful use (22% vs. 13%). The prevalence of medium alcohol use was similar for men and women (9%).

### Prevalence of alcohol use by HIV status and gender

Overall, people living with HIV were more likely to report alcohol use compared to people without HIV (51% v. 67%) and more likely to report low (22% vs. 16%), medium (15% vs. 9%),

**Table 1. Baseline participant characteristics in the PATH/Ekkubo study in Central Uganda (November 2017–March 2020) stratified by gender.**

|  | Total | Men | Women |
|---|---|---|---|
|  | N = 18,460 | n = 7,268 | n = 11,192 |
| **Age (years)** | *N (col %)* | *n (col %)* | *n (col %)* |
| 15–24 | 6,610 (36%) | 2,430 (33%) | 4,180 (37%) |
| 25–34 | 5,492 (30%) | 2,121 (29%) | 3,370 (30%) |
| 35+ | 6,359 (34%) | 2,717 (37%) | 3,642 (33%) |
| **Marital Status** |  |  |  |
| Never married | 3,640 (20%) | 2,248 (31%) | 1,392 (12%) |
| Divorced | 2,626 (14%) | 813 (11%) | 1,813 (16%) |
| Widowed | 482 (3%) | 45 (1%) | 437 (4%) |
| Married | 11,713 (63%) | 4,162 (57%) | 7,550 (67%) |
| **Education** |  |  |  |
| Primary or less | 12,386 (67%) | 4,974 (68%) | 7,411 (66%) |
| Secondary or more | 6,075 (33%) | 2,294 (32%) | 3,781 (34%) |
| **Occupation** |  |  |  |
| Peasant farmer | 7,573 (41%) | 3,072 (42%) | 4,500 (40%) |
| Fisherman | 424 (2%) | 414 (6%) | 10 (.08%) |
| Salaried/business | 2,909 (16%) | 1,301 (18%) | 1,608 (14%) |
| Restaurant/Bar | 489 (3%) | 24 (.33%) | 465 (4%) |
| Market vendor | 362 (2%) | 46 (1%) | 316 (3%) |
| Casual | 1,726 (9%) | 1,046 (14%) | 680 (6%) |
| Not employed | 1,509 (8%) | 295 (4%) | 1,214 (11%) |
| Other | 3,469 (19%) | 1,070 (15%) | 2,399 (21%) |
| **Wealth Index‡** |  |  |  |
| 1 | 6,251 (34%) | 2,575 (35%) | 3,675 (33%) |
| 2 | 4,806 (26%) | 1,884 (26%) | 2,922 (26%) |
| 3 | 2,393 (13%) | 997 (14%) | 1,396 (12%) |
| 4 | 3,025 (16%) | 1,117 (15%) | 1,908 (17%) |
| 5 | 1,986 (11%) | 695 (10%) | 1,291 (12%) |
| **Religion** |  |  |  |
| Protestant | 3,912 (21%) | 1,625 (22%) | 2,286 (20%) |
| Catholic | 8,507 (46%) | 3,421 (47%) | 5,086 (45%) |
| Muslim | 3,787 (21%) | 1,490 (20%) | 2,297 (21%) |
| Other± | 2,255 (12%) | 732 (10%) | 1,523 (14%) |
| **Tribe** |  |  |  |
| Muganda | 11,640 (63%) | 4,700 (65%) | 6,940 (62%) |
| Munyarwanda | 2,947 (16%) | 956 (13%) | 1,990 (18%) |
| Munyoro | 227 (1%) | 67 (1%) | 160 (1%) |
| Murundi | 452 (2%) | 227 (3%) | 225 (2%) |
| Mukiga | 385 (2%) | 177 (2%) | 208 (2%) |
| Mutooro | 108 (1%) | 31 (.43%) | 77 (1%) |
| Munyankole | 1,519 (8%) | 561 (8%) | 958 (9%) |
| Other | 1,183 (6%) | 549 (8%) | 634 (6%) |
| **Depression (CES-D-10)†** |  |  |  |
| <14 | 16,460 (89%) | 6,578 (91%) | 9,881 (88%) |
| ≥14 | 2,001 (11%) | 690 (9%) | 1,311 (12%) |
| **HIV Status** |  |  |  |
| Negative | 17,501 (95%) | 7,012 (96%) | 10,489 (94%) |

*(Continued)*

**Table 1.** (Continued)

|  | Total | Men | Women |
|---|---|---|---|
|  | **N = 18,460** | **n = 7,268** | **n = 11,192** |
| HIV+, newly diagnosed | 288 (1.6%) | 112 (1.5%) | 176 (1.6%) |
| HIV+, previously diagnosed | 671 (4%) | 144 (2%) | 527 (5%) |
| **Alcohol use levels** |  |  |  |
| None | 12,393 (67%) | 4,308 (59%) | 8,085 (72%) |
| Low | 3,016 (16%) | 1,373 (19%) | 1,643 (15%) |
| Medium | 1,688 (9%) | 664 (9%) | 1,024 (9%) |
| High | 811 (4%) | 504 (7%) | 307 (3%) |
| Very High | 552 (3%) | 419 (6%) | 133 (1%) |
| **Harmful alcohol use** | 3051 (17%) | 1,587 (22%) | 1,464 (13%) |

Notes: Alcohol use levels based on the AUDIT-C score: no alcohol use (score 0), non-hazardous/low (score 1–3 for men; 1–2 for women), hazardous/medium (score 4–5 for men; 3–5 for women), hazardous/high (score 6–7), and hazardous/very-high (score 8–12) use, Harmful alcohol use (score $\leq$4 for men; $\leq$3 for women).

$\pm$ Other category is primarily Saved/Pentecostal.

$\ddagger$ The wealth index is based on household characteristics (e.g. roof and floor materials, and whether the household had electricity, television, or a sofa set); and increasing numbers represent increasing wealth.

[†]CES-D-10 = Center for Epidemiological Studies Depression Scale.

A score of 14 or higher indicates individuals at risk for depression.

high (7% vs. 5%), and very high (5% vs. 3%) alcohol use (Chi-squared p-value<0.0001) and this was true among men and women. Fig 1 displays the prevalence of no, low, medium, high, and very high levels of alcohol use stratified by gender, HIV status, and new and known HIV positive diagnosis with design-adjustment. Across all HIV stratifications, men were more likely to report high and very high levels of drinking compared to women; however, this difference was not statistically significant at the p≤0.05 level among those newly diagnosed. Differences in alcohol use between those newly diagnosed with HIV and those with a known HIV diagnosis were not found among men. However, among women, those who were newly diagnosed were more likely report medium, high, and very high levels of alcohol use, compared to those previously diagnosed with HIV.

## Correlates of harmful alcohol use

Table 2 provides the results of multivariable logistic regression models for correlates of harmful alcohol use stratified by men and women. Older age, being divorced, widowed, or married (compared to never married); being a fisherman or market vender (compared to being a peasant farmer) increased the odds of harmful alcohol use among both men and women. Being a member of the Muslim or other religions (compared to Protestant) decreased the odds of harmful alcohol use among both men and women. Increasing wealth was associated with harmful alcohol use; however, there was no trend or dose response.

Among women, having a higher level of education decreased the odds of drinking, while working in a restaurant or bar (compared to peasant farmer) increased the odds of all levels of alcohol use. Compared to being HIV-negative, being newly diagnosed with HIV was associated with increased odds for harmful alcohol use among women, but not men. Further, being previously diagnosed was not associated with harmful alcohol use among either women or men.

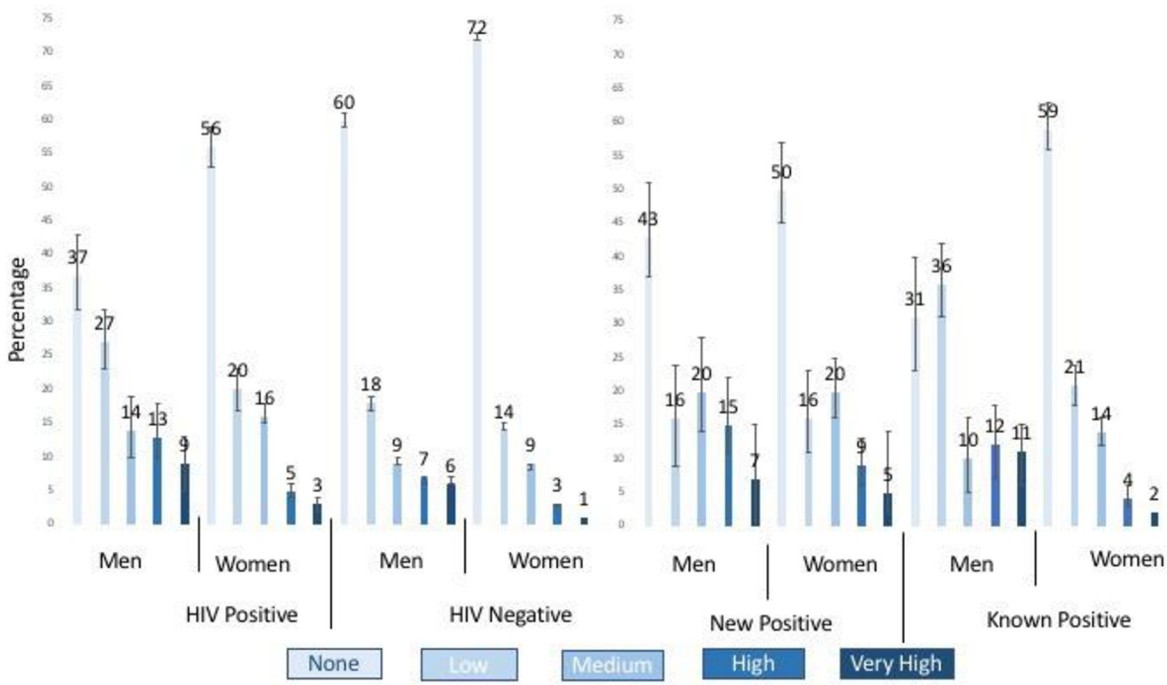

| | HIV positive   n=959 | | HIV negative n=17,501 | | New HIV positive n=288 | | Known HIV positive n=671 | |
|---|---|---|---|---|---|---|---|---|
| | Men n=256 | Women n=703 | Men n=7,012 | Women n=10,489 | Men n=112 | Women n=176 | Men n=144 | Women n=527 |
| No alcohol | 37% (31-43) | 56% (52-60) | 60% (59-61) | 72% (72-73) | 43% (34-53) | 50% (43-58) | 31% (28-40) | 59% (54-63) |
| Low | 27% (22-33) | 20% (17-23) | 18% (17-19) | 14% (14-15) | 16% (10-24) | 16% (11-23) | 36% (28-45) | 21% (18-25) |
| Medium | 14% (10-19) | 16% (13-19) | 9% (8-10) | 9% (9-10) | 20% (13-28) | 20% (15-27) | 10% (5-16) | 14% (11-17) |
| High | 13% (9-18) | 5% (4-7) | 7% (7-8) | 3% (2-3) | 15% (9-23) | 9% (5-14) | 12% (7-18) | 4% (2-6) |
| Very high | 9% (6-14) | 3% (2-4) | 6% (5-6) | 1% (0.9-1) | 7% (3-14) | 5% (2-9) | 11% (6-17) | 2% (1-4) |
| Notes: Alcohol use levels based on the AUDIT-C score: no alcohol use (score 0), low (score 1-3 for men; 1-2 for women), medium (score 4-5 for men; 3-5 for women), high (score 6-7), and very-high (score 8-12) use | | | | | | | | |

**Fig 1. Point estimates and 95% confidence intervals of the prevalence of alcohol use levels (none, low, medium, high, very high, as based on AUDIT-C score) by HIV status and gender from a population-based sample in rural, central Uganda.** Alcohol use levels based on the AUDIT-C score: no alcohol use (score 0), low (score 1–3 for men; 1–2 for women), medium (score 4–5 for men; 3–5 for women), high (score 6–7), and very-high (score 8–12) use.

## Discussion

This study assessed the prevalence of alcohol use with a multi-level measure of consumption stratified by gender and HIV status in the general population in rural, Central Uganda. Using a large sample, our findings strengthen the limited existing literature on alcohol use in this region. Our finding that 33% of participants drank alcohol is similar to the WHO 2018 Global Status report national estimates (36% for current alcohol use in Uganda); in line with the WHO report, we also found that men were more likely to consume alcohol than women [1]. In our sample, people living with HIV were more likely to report low, medium, high and very high levels of alcohol use, compared to those who were HIV negative. Among women, being newly diagnosed with HIV increased the odds of harmful alcohol use (compared to being HIV negative); however, this association was not significant among men.

Few recent studies in Uganda have estimated the prevalence of alcohol use levels by both gender and HIV status (further stratified by newly and previously diagnosed). The most recent

**Table 2. Results from multivariable logistic regression models assessing the relationship between HIV status and harmful alcohol use among women (n = 11,192) and men (n = 7,268) from a population-based sample in rural, central Uganda, June 2017 to March 2020.** (Harmful alcohol use defined as AUDIT-C score ≥3 for women, ≥4 for men).

| | Harmful Alcohol Use | |
|---|---|---|
| | Women OR [95% CI] | Men OR [95% CI] |
| **Age (years)** | | |
| 14–24 | *Referent* | *Referent* |
| 25–34 | 1.53*** [1.30,1.83] | 1.96*** [1.59,2.41] |
| 35+ | 2.09*** [1.67,2.63] | 3.64*** [2.84,4.67] |
| **Marital status** | | |
| Never married | *Referent* | *Referent* |
| Divorced | 1.72*** [1.25,2.35] | 2.03*** [1.55,2.66] |
| Widowed | 1.77** [1.27,2.46] | 2.15* [1.07,4.33] |
| Married | 1.43* [1.12,1.82] | 1.26* [1.01,1.57] |
| **Education** | | |
| None/Primary | *Referent* | *Referent* |
| Secondary/Tertiary | 0.71*** [0.61,0.83] | 0.95 [0.82,1.10] |
| **Religion** | | |
| Protestant | *Referent* | *Referent* |
| Catholic | 1.08 [0.92,1.26] | 0.98 [0.84,1.15] |
| Muslim | 0.24*** [0.19,0.29] | 0.18*** [0.12,0.26] |
| Other | 0.24*** [0.18,0.33] | 0.34*** [0.26,0.46] |
| **Occupation** | | |
| Peasant farmer | *Referent* | *Referent* |
| Fishing industry | 2.85* [1.02,7.99] | 2.17*** [1.57,3.00] |
| Salaried/Business | 1.22 [0.95,1.58] | 0.89 [0.71,1.11] |
| Restaurant/Bar | 4.32*** [3.40,5.51] | 2.75 [0.81,6.31] |
| Market vendor | 2.60*** [2.04,3.31] | 2.75** [1.50,5.04] |
| Casual | 1.85*** [1.48,2.30] | 1.58*** [1.22,2.04] |
| Not employed | 1.05 [0.82,1.33] | 0.58 [0.32,1.05] |
| Other | 1.01 [0.78,1.29] | 1.20 [0.99,1.46] |
| **Wealth Index‡** | | |
| 1 | *Referent* | *Referent* |
| 2 | 0.75*** [0.63,0.89] | 0.81 [0.66,1.01] |
| 3 | 0.92 [0.76,1.12] | 0.90 [0.65,1.25] |
| 4 | 0.73*** [0.59,0.89] | 0.62*** [0.46,0.83] |
| 5 | .98 [0.77,1.25] | 0.64*** [0.46,0.89] |
| **Depression symptoms (CES-D-10 score) †** | | |
| ≤14 | *Referent* | *Referent* |
| >14 | 1.20 [0.96,1.49] | 1.18 [0.81,1.73] |
| **HIV Status** | | |
| Negative | *Referent* | *Referent* |
| Newly diagnosed HIV+ | 2.81*** [2.01,3.91] | 1.42 [0.98,2.05] |

(*Continued*)

**Table 2.** (Continued)

| | Harmful Alcohol Use | |
| --- | --- | --- |
| | **Women OR [95% CI]** | **Men OR [95% CI]** |
| Previously diagnosed HIV+ | 1.03 [.77,1.38] | 0.83 [0.60,1.15] |

Notes: Harmful alcohol use defined as AUDIT-C score ≥4 for men and ≥3 for women (combines medium, high, and very high levels of alcohol use). RRR = relative risk ration, CI = Confidence interval).

‡The wealth index is based on household characteristics (e.g. roof and floor materials, and whether the household had electricity, television, or a sofa set); and increasing numbers represent increasing wealth.

†CES-D-10 = Center for Epidemiological Studies Depression Scale.

A score of 14 or higher indicates individuals at risk for depression. CES-D-10: Center for Epidemiological Studies of Depression Short Form.

*** represents a p-value of <0.001.

WHO global status report on alcohol utilized data from 2016 and did not report on HIV status [1]. An article by Wagman et al. estimated alcohol use among participants in the in Rakai Community Cohort Study (2015–2016) and found that the prevalence of past year alcohol use was 45%, men drank more than women (52% vs. 39%), and HIV status was not associated with alcohol use in adjusted models [13]. However, that study did not stratify by both gender and HIV status, and did not measure alcohol consumption frequency or quantity. Another study that took place in fishing villages in the Wakiso District, found that alcohol use was higher among men compared to women (22.7% vs. 17.9%); however, this study did not stratify by HIV status [27]. Another study among men and women living with HIV in Uganda, Russia, and the United States utilized the AUDIT-C and phosphatidylethanol (PEth), a biomarker of alcohol consumption, to identify gender differences in heavy drinking [10]. This study found that, in Uganda, men were more likely to drink (52%) than women (36%). However, this study purposively enrolled participants who reported heavy drinking and results cannot be used to estimate prevalence. Thus, our research provides point estimates and confidence intervals of alcohol use by both gender and HIV status that can be utilized by researchers as part of systematic reviews and modeling studies, and by policy-makers to determine the burden of alcohol use in important sub-sets of the population.

Our finding that men were more likely to drink alcohol than women in Uganda is consistent with previous research and likely tied to traditional gender roles and spousal expectations. (45) One review found that women experience different biological impacts from alcohol use than men and have different motives for drinking.(5) For example, women were more likely than men to experience "unpleasant emotions" and use alcohol to change negative moods. (5) Qualitative work in Uganda found that it is socially acceptable for men to drink recreationally with friends, and alcohol use was perceived to enhance masculinity.(46) In contrast, women who drink in public areas or in higher quantities could incur certain risks, such as vulnerability to sexual aggression by men.(47) Studies in Rakai, Uganda found that HIV prevalence was higher among women who reported alcohol use before sex (aOR: 1.45; 95% CI: 1.06–1.98) and the association was stronger when women reported both using alcohol before sex and experiencing sexual coercion (OR: 1.79; 95% CI: 1.25–2.56).(48) The SEARCH trial in rural Uganda and Kenya found the highest incidence of HIV was among women who used alcohol, although this relationship was not found among men.(11) Thus, alcohol use may differentially increase the risk for HIV among women compared to men.

While men drink at higher levels, women are more likely to face syndemic conditions, including HIV, alcohol use, depressive symptoms, and intimate partner violence [9]. Research

supports a co-occurring and bidirectional relationship between depression and alcohol use [28]. For example, depression and alcohol use may have common genetic or environmental underlying causes [29]. Alcohol use may be utilized as a tool to deal with negative emotions [5,30]; however, the effects of alcohol use may increase the risk for depression [28]. A study that assessed correlates and effects of alcohol use expectancies among people living with HIV in Uganda found that women with affective depressive symptoms had higher alcohol use expectancies, making them more vulnerable to unhealthy alcohol use [31]. In our study, the covariate, depressive symptoms, and the interaction between depressive symptoms and HIV status were not associated with harmful alcohol use. However, our multivariable model was developed to look at the relationship between HIV status and alcohol use and thus interpreting the effect estimate for depressive symptoms may lead to "Table 2 Fallacy [32]."

We also found that alcohol use was higher in women who had a new HIV diagnosis (HIV test results were provided on the same day as the interview) compared to those who had been previously diagnosed. While our sample size was small for this group, our finding supports previous research that alcohol use is associated with a lower likelihood of being diagnosed with HIV [33]. A study among female sex workers in Malawi found that harmful drinking and alcohol use disorders were more prevalent among women who were unaware of their HIV-positive status compared to those who had been previously diagnosed.(42) A study among people living with HIV in three fishing communities in Uganda found that those with co-occurring hazardous consumption and alcohol-related problems had greater odds of unknown HIV status (adjOR 3.35, 95% CI 1.52–7.42), compared to non-drinkers [34]. While HIV testing is widely available in Uganda [12], increased services may be needed to ensure individuals who consume harmful amounts of alcohol, particularly women, receive HIV testing, linkage to care, and ART initiation.

Our study does not address the directionality of the association between alcohol use and HIV. The higher prevalence of alcohol use among people living with HIV, compared to those without HIV, may reflect the impact of alcohol use on increasing HIV risk. Several systematic reviews and meta-analyses [35–39] as well as event level studies [40,41] report that alcohol use can increase HIV risk by impacting decision-making, resulting in riskier sexual behaviors. A study conducted in fishing communities around Lake Victoria found that alcohol use was associated with incident HIV infections among both men and women [42], and the population attributable fraction of incident HIV infections due to alcohol was 64.1% (95% CI: 23.5–83.1%) [42]. Further, research has shown that alcohol use can contribute to a syndemic with depressive symptoms and intimate partner violence that collectively elevate HIV risk [9]. However, the higher prevalence of alcohol use among people living with HIV may also reflect the use of alcohol as a coping mechanism for the psychological impact of an HIV diagnosis [5,28,30]. A recent study in Uganda, which used both the AUDIT-C and a PEth biomarker, found that individuals living with HIV initially decreased unhealthy alcohol use prior to ART initiation; however, alcohol use increased over time and returned to high levels [43]. Thus, in order to better address alcohol use among people living with HIV in Uganda, more research is needed to understand the drivers and characteristics of alcohol use over time.

Interventions to reduce alcohol use have been implemented in sub-Saharan Africa with mixed effectiveness. Recent systematic review and meta-analysis of non-pharmacological interventions in sub-Saharan Africa found that psychosocial interventions increased alcohol abstinence at 3–6 months and 12–60 months, but had no statistically significant impact on AUDIT scores, drinks per drinking day or percentage of drinking days [44]. While structural interventions, such as reducing the availability of alcohol, increasing prices through taxes, and restricting alcohol advertising have been found to be effective at reducing alcohol use in other settings [45], research is limited in sub-Saharan Africa [44]. Further, given the overlapping

burdens of HIV and alcohol use found among women, there is lack of research on interventions that are gender-responsive and address women's specific needs [46].

Our study estimated the prevalence of alcohol use with a nuanced measure of consumption among a large sample; however, it has several limitations. First, the alcohol use measure is based on self-report, which may have resulted in underreporting of true behaviors as a result of social desirability and recall bias, and under-reporting may vary by sex [47]. Second, non-standard drink sizes and alcohol concentrations may cause misclassification of harmful alcohol use and binge drinking. However, the AUDIT-C metric is a common, validated measure [17] that has been widely used in sub-Saharan Africa, which allows for comparison across multiple settings. Third, some previous studies have found that the social environment, such as the alcohol intake of family members, friends, and partners; and intimate partner violence are important risk factors for harmful alcohol use [48]; however, these variables were not available for analysis and should be considered for future research. Further, alcohol use questions were implemented in the PATH/Ekkubo study in November 2017 and participants enrolled prior were not included in this analysis, thus the results may not be generalizable to all of rural, central Uganda. Finally, this analysis used cross-sectional data and we were unable to assess causation between harmful alcohol use and correlates.

This study estimated the prevalence of alcohol use levels stratified by gender and HIV status in rural, central Uganda. We found that people living with HIV were more likely to report alcohol use, compared to those who were HIV negative. In adjusted models stratified by gender, being newly diagnosed with HIV was associated with harmful alcohol use among women, but not among men. More research is needed to understand how alcohol use may increase the risks of HIV acquisition among women and to identify gender-responsive services to address harmful alcohol use and increase access to HIV testing and linkage to care for women who use harmful levels of alcohol.

## Supporting information

**S1 Data.**
(XLSX)

## Author Contributions

**Conceptualization:** Rhoda K. Wanyenze, Susan M. Kiene.

**Data curation:** Katherine Schmarje Crockett.

**Formal analysis:** Adriane Wynn, Katelyn M. Sileo, Natasha K. Martin, Susan M. Kiene.

**Funding acquisition:** Rhoda K. Wanyenze, Susan M. Kiene.

**Investigation:** Rose Naigino, Michael Ediau, Rhoda K. Wanyenze, Susan M. Kiene.

**Methodology:** Adriane Wynn, Katherine Schmarje Crockett, Susan M. Kiene.

**Resources:** Susan M. Kiene.

**Writing – original draft:** Adriane Wynn.

**Writing – review & editing:** Adriane Wynn, Katelyn M. Sileo, Katherine Schmarje Crockett, Rose Naigino, Michael Ediau, Rhoda K. Wanyenze, Noah Kiwanuka, Natasha K. Martin, Susan M. Kiene.

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
