## [Decision Letter · Decision Letter 0]

20 Jul 2023

PONE-D-23-08339Prevalence of alcohol use by gender and HIV status in rural Uganda.PLOS ONE

Dear Dr. Wynn,

Thank you for submitting your manuscript to PLOS ONE. After careful consideration, we feel that it has merit but does not fully meet PLOS ONE’s publication criteria as it currently stands. Therefore, we invite you to submit a revised version of the manuscript that addresses the points raised during the review process.

We look forward to receiving your revised manuscript.

Kind regards,

Daniel Semakula, M.D. MPH, PhD

Academic Editor

PLOS ONE

Journal Requirements:

2. Please expand the acronym “NIAAA and NIMH” (as indicated in your financial disclosure) so that it states the name of your funders in full.

5. Please amend your manuscript to include your abstract after the title page.

Reviewers' comments:

Reviewer's Responses to Questions

**Comments to the Author**

1. Is the manuscript technically sound, and do the data support the conclusions?

Reviewer #1: Partly

Reviewer #2: Yes

2. Has the statistical analysis been performed appropriately and rigorously? 

Reviewer #1: Yes

Reviewer #2: Yes

3. Have the authors made all data underlying the findings in their manuscript fully available?

Reviewer #1: Yes

Reviewer #2: Yes

4. Is the manuscript presented in an intelligible fashion and written in standard English?

Reviewer #1: Yes

Reviewer #2: Yes

5. Review Comments to the Author

Reviewer #1: 1. General comments

• To me, the why of this study is not clear at all. Position this paper as a mental health related paper. It makes sense that way, otherwise, the why is not clear.

2. Methods

• This section is difficult to read because of how its arranged. It would be helpful to have sub headings with outcomes, exposure variables and confounding factors. It’s easier to follow like that

3. Results

• Because there are so many components to the analysis, it would be useful to also have sub sections here. As it is, it is really difficult to go through the results section as you have to figure out on your own as a reader where the demographic characteristics ends and regression results are starting.

4. Discussion

• My understanding of the analysis was that the authors are trying to understand the role of gender and HIV status in alcohol consumption. But if you look at the studies that they are referencing, mostly focuses on alcohol as a risk factor for HIV infection. I was expecting the discussion to be more about the mental health related reasons that could drive those who have newly been diagnosed with HIV to get into drinking. Similarly, the cultural expectations/norms (which the authors have briefly talked about) to be more prevalent in the discussion about alcohol use and gender. Cite the studies that have looked at gender and alcohol use. The same thing with HIV diagnosis and alcohol consumption, cite more of such studies.

• And then the authors can also briefly look at the existing evidence which shows alcohol use as a risk factor for HIV (do not make this the focus of the discussion as it is currently, because that’s not what your study focused on).

5. Study implication

• What are the implications of these findings? Why should policy makers care about this study at all? How does this link to the policy direction on alcohol use, mental health, and HIV care in Uganda?

Reviewer #2: In this population level cross-sectional study, the authors assessed prevalence of alcohol use by gender and HIV status in rural Uganda. They found that 33% prevalence of alcohol use with higher alcohol consumption in men compared to women.

The manuscript does not include line numbers making it difficult to pinpoint the comments to the specific line.

The paper is well written and largely coherent. However, the authors need to provide the exact original contribution that the manuscript is adding to the literature. As they have discussed, the findings have been previously demonstrated. It would be helpul if the authors attempted to clearly state what new information this paper is adding and how it differs from the many that have been previously published on the topic of gender and alcohol.

Abstract: In the results section of the abstract ….”Men were less likely to report no alcohol use , and more likely …” Please rephrase sentence with alcohol use rather than the “no alcohol use”

Include succinct research or programmatic recommendation in the abstract

Introduction:

Paragraph three “Alcohol use and related outcomes vary by gender “ --- the authors should state the exact outcomes of interest .

The authors should include the expected or anticipated benefits of this study i.e estimating alcohol use by gender so that the interest of the readers are captured at the outset.

Methods :

Third Last line on page 3 : “ Thes levels have been identified …..and adverse clinical outcomes” . The authors should state the outcomes or examples of these outcomes.

Page 4 : although authors mention how information of estimated units of alcohol were captured from participants … they do not mention exactly how these were used in filling the AUDIT-C form or how estimated alcohol was arrived at.

On HIV testing : the authors used the term “new positive’ and “new negative” . Negative is negative and there is no term as new negative. Wouldn’t the terms “newly diagnosed HIV positive “ and “HIV negative” communicate better? Was HIV recency testing done ?

Results:

The abstract says age range is 15-59 yrs but tables include 14 years -please resolve this consistency .

The tables indicate that age range started at 14 years , The authors need to clarify why they included minors and why not use 18 yrs and above only sample ?

The results do not include assessment of intimate partner violence , was this data collected ? what were the results of IPV and HIV results in this population?

Discussion

Page 8 paragraph 2 : On the directionality of alcohol use and HIV: what happens when a person consuming alcohol is diagnosed with HIV ? Does alcohol consumption reduce after an HIV diagnosis?

Page 8 last paragraph: “We found that newly diagnosed with HIV was associated with harmful ……..” Taken together what does this differential HIV risk by gender mean? Are there any recommendations for prevention?

Discuss the HIV recency testing results if available or not available

The discussion calls for further research, but the statements are generic. Please provide a more nuanced recommendation for the further research that is needed. Also provide Programmatic recommendations if any

6. PLOS authors have the option to publish the peer review history of their article (what does this mean?). If published, this will include your full peer review and any attached files.

Reviewer #1: No

Reviewer #2: No

---

## [Author Response · Author response to Decision Letter 0]

13 Sep 2023

A word copy is attached with the files:

Re: Response to reviewers, PONE-D-23-08339

Dear Reviewers:

Thank you for taking the time to review our manuscript entitled, “Prevalence of alcohol use by gender and HIV status in rural Uganda.” Your comments and suggestions have been very valuable, and we believe have improved the manuscript. Please see our response to each comment below.

Reviewer #1

1. General comments

• To me, the why of this study is not clear at all. Position this paper as a mental health related paper. It makes sense that way, otherwise, the why is not clear.

Response: Thank you for encouraging us to clarify the overall goals of the paper. We believe this research is among a very limited number of studies that utilize a large and recent sample in Uganda to estimate the prevalence of alcohol use stratified by both HIV status and gender. We have clarified the goals in the introduction (lines 72-73): “Such nuanced prevalence estimates are needed by researchers and policy-makers to understand the burden of alcohol use and to determine service needs.” We also made significant changes to the discussion. Paragraph two (lines 226-235) clarifies the gaps in the literature that our study fills. For example, we discuss recent studies in Uganda that stratify by HIV or gender, but not both.

Thank you for bringing up the important point about the relationship between mental health, alcohol use and HIV. In paragraph four of the discussion (lines 255-265), we have added content on syndemic conditions that are particularly common among women, including depression. Alcohol use and depression co-occur and there is evidence supporting a bidirectional relationship, with each serving as both risk factor and sequelae of the other (Boden and Fergusson 2011). The exact nature of this relationship has been difficult to disentangle, but the strong association has been explained through several pathways: depression and alcohol use may have common genetic or environmental underlying causes (Kendler, Prescott et al. 2003), depression may increase risk of alcohol use (self-medication model) (Bolton, Robinson et al. 2009), or alcohol use may increase risk of subsequent depression (as a depressant and due to consequences of one’s drinking) (Fergusson, Boden et al. 2009). We conclude the paragraph with: “In our study, the covariate, depressive symptoms, and the interaction between depressive symptoms and HIV status were not associated with harmful alcohol use. However, our multivariable model was developed to look at the relationship between HIV status and alcohol use and thus interpreting the effect estimate for depressive symptoms may lead to “Table 2 Fallacy.”(52)”

2. Methods

• This section is difficult to read because of how its arranged. It would be helpful to have sub-headings with outcomes, exposure variables and confounding factors. It’s easier to follow like that.

Response: As suggested, to improve clarity, we have added the following sub-headings to the methods section: Outcome measure: Alcohol use; Main exposures: HIV status and gender; and Other correlates of harmful alcohol use. 

3. Results

• Because there are so many components to the analysis, it would be useful to also have sub sections here. As it is, it is really difficult to go through the results section as you have to figure out on your own as a reader where the demographic characteristics ends and regression results are starting.

Response: As suggested, to improve clarity, we have added the following sub-headings to the results section: Sample characteristics; Prevalence of alcohol use by gender; Prevalence of alcohol use by HIV status and gender; and correlates of harmful alcohol use

4. Discussion

• My understanding of the analysis was that the authors are trying to understand the role of gender and HIV status in alcohol consumption. But if you look at the studies that they are referencing, mostly focuses on alcohol as a risk factor for HIV infection. I was expecting the discussion to be more about the mental health related reasons that could drive those who have newly been diagnosed with HIV to get into drinking. Similarly, the cultural expectations/norms (which the authors have briefly talked about) to be more prevalent in the discussion about alcohol use and gender. Cite the studies that have looked at gender and alcohol use. The same thing with HIV diagnosis and alcohol consumption, cite more of such studies. And then the authors can also briefly look at the existing evidence which shows alcohol use as a risk factor for HIV (do not make this the focus of the discussion as it is currently, because that’s not what your study focused on).

Response: We agree with the reviewer comments and have added new content that discusses the role of gender and HIV status on alcohol consumption (lines 242-254): “Our finding that men were more likely to drink alcohol than women in Uganda is consistent with previous research and likely tied to traditional gender roles and spousal expectations.(45) One review found that women experience different biological impacts from alcohol use than men and have different motives for drinking.(5) For example, women were more likely than men to experience “unpleasant emotions” and use alcohol to change negative moods. Qualitative work in Uganda found that it is socially acceptable for men to drink recreationally with friends, and alcohol use was perceived to enhance masculinity.(46) In contrast, women who drink in public areas or in higher quantities could incur certain risks, such as vulnerability to sexual aggression by men.(47) Studies in Rakai, Uganda found that HIV prevalence was higher among women who reported alcohol use before sex (aOR: 1.45; 95% CI: 1.06-1.98) and the association was stronger when women reported both using alcohol before sex and experiencing sexual coercion (OR: 1.79; 95% CI: 1.25-2.56).(48) The SEARCH trial in rural Uganda and Kenya found the highest incidence of HIV was among women who used alcohol, although this relationship was not found among men.(11) Thus, alcohol use may differentially increase the risk for HIV among women compared to men.” 

As discussed above, we also added more discussion about symdemics and depression, particularly among women.

5. Study implication

• What are the implications of these findings? Why should policy makers care about this study at all? How does this link to the policy direction on alcohol use, mental health, and HIV care in Uganda?

Response: Thank you for asking us to clarify the big picture implications of this study. We believe that this is a relatively simple analysis that provides previously unavailable point-estimates and confidence intervals of the prevalence of alcohol use levels stratified by both gender and HIV. These estimates can be used by researchers and policy-makers to better understand the burden of alcohol use by important sub-groups in Uganda, and the potential scale of services needed (lines 239-241). The results can also be used by modelers as parameters to estimate the impact of interventions to address harmful alcohol. Further, the finding that HIV status may have a differential relationship with alcohol use by gender demonstrates a need for more research (lines 315-318) “to understand how alcohol use may increase the risks of HIV acquisition among women and to identify gender-responsive services to address harmful alcohol use and increase access to HIV testing and linkage to care for women who use harmful levels of alcohol.”

Reviewer #2

1. General Comments

• The manuscript does not include line numbers making it difficult to pinpoint the comments to the specific line.

The paper is well written and largely coherent. However, the authors need to provide the exact original contribution that the manuscript is adding to the literature. As they have discussed, the findings have been previously demonstrated. It would be helpul if the authors attempted to clearly state what new information this paper is adding and how it differs from the many that have been previously published on the topic of gender and alcohol.

Response: We have added line numbers. 

2. Abstract 

• In the results section of the abstract ….”Men were less likely to report no alcohol use , and more likely …” Please rephrase sentence with alcohol use rather than the “no alcohol use”

Response: Agreed. We have updated the sentence: “Compared to women, men were more likely to report alcohol use (Chi-squared p-value<0.0001).”

• Include succinct research or programmatic recommendation in the abstract

Response: We have updated the abstract: “More research is needed to understand how alcohol use may increase the risks of HIV acquisition among women and to identify gender-responsive services to address harmful alcohol use and increase access to HIV testing and linkage to care for women who use harmful levels of alcohol.”

3. Introduction

• The authors should include the expected or anticipated benefits of this study i.e estimating alcohol use by gender so that the interest of the readers are captured at the outset.

Response: We have added the following sentence to the introduction (lines 72-73): “Such nuanced prevalence estimates are needed by researchers and policy-makers to understand the burden of alcohol use and to determine service needs.”

4. Methods

• Third Last line on page 3 : “ These levels have been identified …..and adverse clinical outcomes” . The authors should state the outcomes or examples of these outcomes.

Response: We have updated the sentence: “These levels have been identified in previous literature as demonstrating increased severity and adverse HIV care cascade outcomes, including reduced likelihood of ART, ART adherence, and viral suppression among populations in the United States and East Africa.(15, 18)”

• Page 4 : although authors mention how information of estimated units of alcohol were captured from participants … they do not mention exactly how these were used in filling the AUDIT-C form or how estimated alcohol was arrived at.

Response: We have tried to clarify by including the following sentence (lines 110-112): “Using the pictures as a guide, participants estimated the number of single drinks they had on a typical day when drinking and how often they consumed six or more single drinks on one occasion.”

• On HIV testing : the authors used the term “new positive’ and “new negative” . Negative is negative and there is no term as new negative. Wouldn’t the terms “newly diagnosed HIV positive “ and “HIV negative” communicate better? Was HIV recency testing done ?

Response: We agree that “new negative” does not make sense and have made sure this is not included in the manuscript. HIV recency testing was not done. 

5. Results:

• The abstract says age range is 15-59 yrs but tables include 14 years -please resolve this consistency .

The tables indicate that age range started at 14 years , The authors need to clarify why they included minors and why not use 18 yrs and above only sample ?

Response: Thank you. 14 years was a typo that has been corrected. The sample included, “adults aged 18 to 59 years old or emancipated minors (e.g. individuals married or with children) aged 15 to 17 years.”

• The results do not include assessment of intimate partner violence , was this data collected ? what were the results of IPV and HIV results in this population?

Response: IPV was not available for this analysis and we have clarified in the limitations section that IPV should be considered for assessment in future research: “Third, some previous studies have found that the social environment, such as the alcohol intake of family members, friends, and partners; and intimate partner violence are important risk factors for harmful alcohol use;(61) however, these variables were not available for analysis and should be considered for future research.”

6. Discussion

• Page 8 paragraph 2 : On the directionality of alcohol use and HIV: what happens when a person consuming alcohol is diagnosed with HIV ? Does alcohol consumption reduce after an HIV diagnosis?

Response: We have added content related to directionality in paragraph six of the discussion, including: “Further, our results may indicate that individuals reduce their alcohol use after being diagnosed with HIV. One study in Uganda found that among a sample of people living with HIV who used alcohol in the past year, most reported attempting to cease or reduce alcohol intake after entering HIV care.(56) However, another study that used both the AUDIT-C and PEth biomarker found that individuals initially decreased unhealthy alcohol use prior to ART initiation; however, alcohol use increased over time and returned to high levels.(54)”

• Page 8 last paragraph: “We found that newly diagnosed with HIV was associated with harmful ……..” Taken together what does this differential HIV risk by gender mean? Are there any recommendations for prevention?

Response: We have updated the conclusion to call for more research to understand how alcohol use may increase the risks of HIV acquisition among women and to identify gender-responsive services to reduce alcohol use and increase HIV testing and linkage to care among women who drink.

• Discuss the HIV recency testing results if available or not available

Response: HIV recency testing was not conducted.

• The discussion calls for further research, but the statements are generic. Please provide a more nuanced recommendation for the further research that is needed. Also provide Programmatic recommendations if any

Response: Based on your comments, we have updated the conclusion: “This study estimated the prevalence of alcohol use levels stratified by gender and HIV status in rural, central Uganda. We found that people living with HIV were more likely to report alcohol use, compared to those who were HIV negative. In adjusted models stratified by gender, being newly diagnosed with HIV was associated with harmful alcohol use among women, but not among men. More research is needed to understand how alcohol use may increase the risks of HIV acquisition among women and to identify gender-responsive services to address harmful alcohol use and increase access to HIV testing and linkage to care for women who use harmful levels of alcohol.”

Thank you for considering our manuscript. We appreciate your time and look forward to your response.

---

## [Decision Letter · Decision Letter 1]

23 Feb 2024

PONE-D-23-08339R1Prevalence of alcohol use by gender and HIV status in rural Uganda.PLOS ONE

Dear Dr. Wynn,

Thank you for submitting your manuscript to PLOS ONE. After careful consideration, we feel that it has merit but does not fully meet PLOS ONE’s publication criteria as it currently stands. Therefore, we invite you to submit a revised version of the manuscript that addresses the points raised during the review process.

We look forward to receiving your revised manuscript.

Kind regards,

Miquel Vall-llosera Camps

Staff Editor

PLOS ONE

Reviewers' comments:

Reviewer's Responses to Questions

**Comments to the Author**

1. If the authors have adequately addressed your comments raised in a previous round of review and you feel that this manuscript is now acceptable for publication, you may indicate that here to bypass the “Comments to the Author” section, enter your conflict of interest statement in the “Confidential to Editor” section, and submit your "Accept" recommendation.

Reviewer #1: (No Response)

Reviewer #3: All comments have been addressed

Reviewer #4: All comments have been addressed

2. Is the manuscript technically sound, and do the data support the conclusions?

Reviewer #1: Partly

Reviewer #3: Yes

Reviewer #4: Yes

3. Has the statistical analysis been performed appropriately and rigorously? 

Reviewer #1: Yes

Reviewer #3: Yes

Reviewer #4: Yes

4. Have the authors made all data underlying the findings in their manuscript fully available?

Reviewer #1: No

Reviewer #3: Yes

Reviewer #4: Yes

5. Is the manuscript presented in an intelligible fashion and written in standard English?

Reviewer #1: Yes

Reviewer #3: Yes

Reviewer #4: Yes

6. Review Comments to the Author

Reviewer #1: I maintain the view that the research question has not been adequately addressed. After reviewing the abstract, results, and discussion sections, it is apparent that while the authors have made efforts to incorporate the gender component, there remains a lack of sufficient exploration into the impact of HIV on alcohol use. There appears to be some ambiguity regarding the direction of the relationship between alcohol abuse and HIV. From my interpretation of the study's objective, it seems the authors aimed to investigate alcohol use based on HIV status. Specifically, do individuals with HIV engage in more alcohol consumption compared to those without the virus? Additionally, does the timing of the diagnosis play a role? It appears to be related to coping mechanisms following an HIV-positive diagnosis. The focus of the study should align more with the existing literature, emphasizing the relationship between alcohol use and the psychological impact of an HIV diagnosis rather than exploring whether alcohol abuse increases the risk of HIV, particularly since the study centers on individuals already diagnosed with the virus.

Reviewer #3: In this manuscript by Adriane Wynn et al, the authors provide alcohol use prevalence from a cross-sectional analysis in central Uganda, stratified by sex and HIV status, and evaluate correlates of alcohol use, and the relationship between alcohol use and HIV diagnosis status (persons newly diagnosed HIV+, previously diagnosed HIV+ and HIV-negative). The manuscript adds to and supports previously published literature. Overall, I found the authors to be highly responsive to the comments provided by the two reviewers (though I was not one of the initial reviewers). I found the justification for the analysis (the "why" requested by Reviewer 1) to be well explained, the Methods were clear (following revisions suggested by both Reviewers), and the authors provide a thorough review of existing literature on the topic (epidemiology of alcohol use in Uganda and its relationship to HIV infection), and explain how their results add to the literature.

One minor point in the Discussion section for the authors to consider, in the paragraph on limitations:

- The authors should consider noting (see lines 341-342) that under-reporting of alcohol use may vary by sex.

Reviewer #4: The revised version of the article has largely responded to the major comments raised earlier. I concur that the article is better pitched as a mental health related paper. I acknowledge the response by the authors however the article would have been stronger as a mental health paper. There remain a few portions that require minor edits:

Line 228- rephrase “less likely to report no alcohol use”

Line 234 include the p-value

Line 249 edit this “associated with increased the likelihood”

Line 326 This study does not have the ability to demonstrate that “alcohol use reduced after diagnosis” due to the study design.

I recommend the article for publication after these minor edits.

7. PLOS authors have the option to publish the peer review history of their article (what does this mean?). If published, this will include your full peer review and any attached files.

Reviewer #1: **Yes: **Martina Mchenga

Reviewer #3: No

Reviewer #4: No

---

## [Author Response · Author response to Decision Letter 1]

10 Apr 2024

We thank reviewers for the additional comments and suggestions. Our responses are outlined below. (Also see the attached cover letter)

Reviewer #1: I maintain the view that the research question has not been adequately addressed. After reviewing the abstract, results, and discussion sections, it is apparent that while the authors have made efforts to incorporate the gender component, there remains a lack of sufficient exploration into the impact of HIV on alcohol use. There appears to be some ambiguity regarding the direction of the relationship between alcohol abuse and HIV. From my interpretation of the study's objective, it seems the authors aimed to investigate alcohol use based on HIV status. Specifically, do individuals with HIV engage in more alcohol consumption compared to those without the virus? Additionally, does the timing of the diagnosis play a role? It appears to be related to coping mechanisms following an HIV-positive diagnosis. The focus of the study should align more with the existing literature, emphasizing the relationship between alcohol use and the psychological impact of an HIV diagnosis rather than exploring whether alcohol abuse increases the risk of HIV, particularly since the study centers on individuals already diagnosed with the virus.

Thank you for your ongoing review of our article. We agree that there continues to be ambiguity regarding the direction of the relationship between harmful alcohol use and HIV diagnosis. Due to the cross-sectional nature of our data, we were unable to disentangle whether previous alcohol use contributed to increased risk of HIV acquisition or whether living with HIV increased alcohol use as a coping mechanism. We have updated paragraph six in our discussion in order to better reflect this uncertainty and further underscore the reviewer comments that alcohol use may be higher among people living with HIV because it is used as a coping mechanism.

“Our study does not address the directionality of the association between alcohol use and HIV. The higher prevalence of alcohol use among people living with HIV, compared to those without HIV, may reflect the impact of alcohol use on increasing HIV risk. Several systematic reviews and meta-analyses(35-39) as well as event level studies (40, 41) report that alcohol use can increase HIV risk by impacting decision-making, resulting in riskier sexual behaviors. A study conducted in fishing communities around Lake Victoria found that alcohol use was associated with incident HIV infections among both men and women,(42) and the population attributable fraction of incident HIV infections due to alcohol was 64.1% (95% CI: 23.5-83.1%).(42) Further, research has shown that alcohol use can contribute to a syndemic with depressive symptoms and intimate partner violence that collectively elevate HIV risk.(9) However, the higher prevalence of alcohol use among people living with HIV may also reflect the use of alcohol as a coping mechanism for the psychological impact of an HIV diagnosis.(5, 28, 30) A recent study in Uganda, which used both the AUDIT-C and a PEth biomarker, found that individuals living with HIV initially decreased unhealthy alcohol use prior to ART initiation; however, alcohol use increased over time and returned to high levels.(43) Thus, in order to better address alcohol use among people living with HIV in Uganda, more research is needed to understand the drivers and characteristics of alcohol use over time.”

Reviewer #3: In this manuscript by Adriane Wynn et al, the authors provide alcohol use prevalence from a cross-sectional analysis in central Uganda, stratified by sex and HIV status, and evaluate correlates of alcohol use, and the relationship between alcohol use and HIV diagnosis status (persons newly diagnosed HIV+, previously diagnosed HIV+ and HIV-negative). The manuscript adds to and supports previously published literature. Overall, I found the authors to be highly responsive to the comments provided by the two reviewers (though I was not one of the initial reviewers). I found the justification for the analysis (the "why" requested by Reviewer 1) to be well explained, the Methods were clear (following revisions suggested by both Reviewers), and the authors provide a thorough review of existing literature on the topic (epidemiology of alcohol use in Uganda and its relationship to HIV infection), and explain how their results add to the literature.

One minor point in the Discussion section for the authors to consider, in the paragraph on limitations:

- The authors should consider noting (see lines 341-342) that under-reporting of alcohol use may vary by sex.

Thank you. We agree and have added this point: “…and under-reporting may vary by sex.”

Reviewer #4: The revised version of the article has largely responded to the major comments raised earlier. I concur that the article is better pitched as a mental health related paper. I acknowledge the response by the authors however the article would have been stronger as a mental health paper. There remain a few portions that require minor edits:

Line 228- rephrase “less likely to report no alcohol use”

We edited as follows: Compared to men, women were more likely to abstain from alcohol use (59% vs. 72%)

Line 234 include the p-value

We removed this typo. The p-value associated with the Chi-squared test was <0.0001.

Line 249 edit this “associated with increased the likelihood”

We edited this sentence to read: “Increasing wealth was associated with harmful alcohol use; however, there was no trend or dose response.”

Line 326 This study does not have the ability to demonstrate that “alcohol use reduced after diagnosis” due to the study design.

We have removed this sentence and added a paragraph that seeks to address Reviewer 1’s continued concerns:

---

## [Decision Letter · Decision Letter 2]

3 May 2024

Prevalence of alcohol use by gender and HIV status in rural Uganda.

PONE-D-23-08339R2

Dear Dr. Adriane,

We’re pleased to inform you that your manuscript has been judged scientifically suitable for publication and will be formally accepted for publication once it meets all outstanding technical requirements.

Kind regards,

Kahsu Gebrekidan

Academic Editor

PLOS ONE

Additional Editor Comments (optional):

Reviewers' comments:

Reviewer's Responses to Questions

**Comments to the Author**

1. If the authors have adequately addressed your comments raised in a previous round of review and you feel that this manuscript is now acceptable for publication, you may indicate that here to bypass the “Comments to the Author” section, enter your conflict of interest statement in the “Confidential to Editor” section, and submit your "Accept" recommendation.

Reviewer #4: (No Response)

2. Is the manuscript technically sound, and do the data support the conclusions?

Reviewer #4: (No Response)

3. Has the statistical analysis been performed appropriately and rigorously? 

Reviewer #4: (No Response)

4. Have the authors made all data underlying the findings in their manuscript fully available?

Reviewer #4: (No Response)

5. Is the manuscript presented in an intelligible fashion and written in standard English?

Reviewer #4: (No Response)

6. Review Comments to the Author

Reviewer #4: (No Response)

7. PLOS authors have the option to publish the peer review history of their article (what does this mean?). If published, this will include your full peer review and any attached files.

Reviewer #4: No

---

## [Editor Report · Acceptance letter]

13 Jun 2024

PONE-D-23-08339R2 

PLOS ONE

Dear Dr. Wynn, 

I'm pleased to inform you that your manuscript has been deemed suitable for publication in PLOS ONE. Congratulations! Your manuscript is now being handed over to our production team.

Kind regards, 

on behalf of

Dr. Kahsu Gebrekidan 

Academic Editor

PLOS ONE